# AtZAT4, a C_2_H_2_-Type Zinc Finger Transcription Factor from *Arabidopsis thaliana*, Is Involved in Pollen and Seed Development

**DOI:** 10.3390/plants11151974

**Published:** 2022-07-29

**Authors:** A. Carolina Puentes-Romero, Sebastián A. González, Enrique González-Villanueva, Carlos R. Figueroa, Simón Ruiz-Lara

**Affiliations:** 1Laboratorio de Genómica Funcional, Institute of Biological Sciences, Universidad de Talca, Talca 3460000, Chile; apuentes@utalca.cl (A.C.P.-R.); sgonzalez@utalca.cl (S.A.G.); egonzale@utalca.cl (E.G.-V.); 2Millenium Nucleus for the Development of Super Adaptable Plants (MN-SAP), Santiago 8340755, Chile; cfigueroa@utalca.cl; 3Laboratory of Plant Molecular Physiology, Institute of Biological Sciences, Universidad de Talca, Talca 3460000, Chile

**Keywords:** pollen development, seed development, zinc-finger proteins, ZAT, *Arabidopsis thaliana*

## Abstract

Pollen plays an essential role in plant fertility by delivering the male gametes to the embryo sac before double fertilization. In several plant species, including Arabidopsis, C_2_H_2_-type zinc-finger transcription factors (TFs) have been involved in different stages of pollen development and maturation. ZINC FINGER of *Arabidopsis* *thaliana* 4 (AtZAT4) is homologous to such TFs and subcellular localization analysis has revealed that AtZAT4 is located in the nucleus. Moreover, analysis of *AtZAT4* expression revealed strong levels of it in flowers and siliques, suggesting a role of the encoded protein in the regulation of genes that are associated with reproductive development. We characterized a T-DNA insertional heterozygous mutant *Atzat4* (+/−). The relative gene expression analysis of *Atzat4* (+/−) showed significant transcript reductions in flowers and siliques. Furthermore, the *Atzat4* (+/−) phenotypic characterization revealed defects in the male germline, showing a reduction in pollen tube germination and elongation. *Atzat4* (+/−) presented reduced fertility, characterized by a smaller silique size compared to the wild type (WT), and a lower number of seeds per silique. Additionally, seeds displayed lower viability and germination. Altogether, our data suggest a role for AtZAT4 in fertilization and seed viability, through the regulation of gene expression associated with reproductive development.

## 1. Introduction

In angiosperm, the pollen grain is the male gametophyte. It plays a major role in plant fertility by generating sperm cells and conveying them to the embryo sac for double fertilization [1]. The correct pollen grain development, the functional pollen tube growth and sperm cells release are necessary for plant sexual reproduction and maintenance of genetic diversity [2]. Microgametophyte development involves the coordinated participation of sporophytic and gametophytic cell types, as well as the gene expressions that control their development in these tissues [3,4]. The Arabidopsis pollen grain development in the anther is well characterized. It occurs in the following three main stages: (i) microsporogenesis (microsporocytes meiosis), (ii) post-meiotic development of microspores (microspores release), and (iii) micro-gametogenesis (mitosis of microspores) [4]. These stages take place inside the anthers on the stamens. The stamen specification, like other floral whorls in *A*. *thaliana*, has been well characterized at the molecular level. According to the floral quartet model, genes encoding for transcription factors (TFs) of the MADS-box type participate in the identification of each flower whorl through quaternary protein complex formation. In this model, stamens are specified by class B+C+E genes [5,6,7].

Following the stamen specification, several TFs regulate specific gene expression within the anther, controlling the male germline development [8]. In this context, several C_2_H_2_-type Zinc-finger proteins (C_2_H_2_-ZFPs) participate in the regulatory network of anther and pollen grain development. To date, 176 genes that encode C_2_H_2_-ZFP have been identified in Arabidopsis, 189 in rice [9,10], 211 in maize [11], and 301 in *Brassica rapa* [12]. Most of the C_2_H_2_-ZFPs that are associated with plant growth, development and environmental stress responses are of Q-type, which present a conserved QALGGH motif in their zinc finger (ZF) domains. These have been exclusively identified in plants, suggesting one or more roles in processes unique to plants [13,14]. Accordingly, these TFs have been associated with specific plant abiotic stresses, including cold and salinity [15]. However, in addition to their role in stress response, several studies have shown their involvement during flower development. For example, SUPERMAN (SUP) is a Q-type C_2_H_2_-ZFP in *Arabidopsis*, with an ethylene-responsive element binding factor-associated amphiphilic repression (EAR) motif, which is essential for maintaining the boundary between stamens and carpels [16,17,18]. In addition, KNUCKLES (KNU) is a small protein containing a single C_2_H_2_ ZF domain with an EAR-like active repression motif, which is essential during flower development for balancing cell proliferation and differentiation. Thus, KNU regulates cellular proliferation in the basal gynoecium tissues [13,19]. 

Genes encoding for C_2_H_2_-ZFPs, which have been described in *Petunia hybrida*, were temporarily and sequentially expressed during anther and pollen development [8]. Among them, TAPETUM DEVELOPMENT ZINC FINGER PROTEIN1 (TAZ1) plays an essential role in post-meiotic tapetum development [20], and MEIOSIS-ASSOCIATED ZINC FINGER PROTEIN 1 (MEZ1) participates in microsporocyte meiosis [21]. In *Arabidopsis*, two C_2_H_2_-ZFPs, DUO POLLEN 1 (DUO1)-ACTIVATED ZINC FINGER 1/2 (DAZ1/DAZ2) have been described in pollen development, specifically during germ cell division. Interestingly, these two TFs also display an EAR transcriptional repression motif [22]. Additionally, in *Arabidopsis*, the C_2_H_2_-ZFP MALE FERTILITY-ASSOCIATED ZINC FINGER PROTEIN 1 (MAZ1) is involved in pollen grain wall development [16]. In *Brassica campestris*, the C_2_H_2_-ZFP BcMF20 has been proposed to be involved in the development of tapetum [23]. Lastly, BrZFP38 from *B. rapa* is involved in the late development of pollen grain as well as pollination [24]. 

While a function of C_2_H_2_ ZFPs has been assigned to pollen development, their involvement in embryo and seed development is less documented. TITAN-LIKE (TTL) is a protein with two atypical C_2_H_2_ motifs. It plays a role in Arabidopsis endosperm development, particularly in endosperm nuclear divisions, and its defective activity leads to defects in the developing embryo [25]. *GmZFP1* from *Glycine max*, that encodes for a C_2_H_2_ ZFP with a single ZF domain with a conserved QALGGH motif, has been associated with late stage seed development. *GmZFP1* was found to be strongly expressed from 45 days after flowering, while the gene expression was lower in the early seed development stage [26]. 

In this study, we characterized ZINC FINGER of *A*. *thaliana* 4 (AtZAT4; AT2G45120) as a C_2_H_2_-ZFP TF involved in reproductive development. AtZAT4 is an uncharacterized C_2_H_2_-ZFP but is associated with a similar function to AtZAT6 and AtZAT10, which are involved in biotic and/or abiotic stress responses [27,28]. The subcellular localization of AtZAT4 is in the nucleus and the transcriptional profile suggests that, in contrast to AtZAT6 and AtZAT10, AtZAT4 could be a regulating gene involved in pollen and seed development. We further characterized a heterozygous insertional mutant of *AtZAT4* [*Atzat4* (+/−)]. The mutant showed defects in the male germline, essentially during pollen germination and in the elongation tube. Accordingly, the fertility of *A. thaliana* was affected with the mutant exhibiting a lower number of developed seeds in comparison to the wild type (WT).

## 2. Results

### 2.1. AtZAT4 Is a Transcription Factor Similar to Other C_2_H_2_-ZFPs Associated with Reproductive Development in Model Plants

To know the evolutionary relationships of AtZAT4 with other C_2_H_2_-type zinc-finger proteins (C_2_H_2_-ZFP) involved in reproductive development, a multiple alignment (Appendix A) and a phylogenetic tree were constructed using the full-length amino acid sequences of AtZAT4 and other sequences of C_2_H_2_-ZFP from *Arabidopsis thaliana, Petunia hybrida*, *Silene latifolia*, *Brassica campestris* ssp. *chinensis* and *Brassica rapa* ssp. *chinensis*. The results showed that AtZAT4 belongs to the same clade as ZPT3-3 from *P. hybrida* [29], and ZPT3-1 from *S. latifolia* [30] (Figure 1). These results suggested that AtZAT4 is a C_2_H_2_-ZF transcription factor (TF) and is evolutionarily closest to TFs associated with reproductive development.

To determine the AtZAT4 subcellular localization, we transiently expressed a carboxyl-terminal fusion of AtZAT4 to the green fluorescent protein (GFP) under the control of CaMV 35S promoter in onion epidermal cells using the bombardment transformation method. The signal of AtZAT4:sGFP was observed only in the nucleus, while the control showed a GFP signal in both the nucleus and cytoplasm. It is important to note that GFP is a small protein that diffuses in a non-specifical way through nuclear pores [31]. The subcellular localization of AtZAT4 was consistent not only with the nuclear localization signal (NLS) in silico prediction (Appendix A), but also with a function of AtZAT4 as a TF (Figure 2).

To identify the putative role of *AtZAT4* in *A. thaliana* development, we studied the organ-specific transcriptional profile of this gene. For this purpose, we proceeded to measure the level of transcripts using reverse transcription-quantitative real-time PCR (RT-qPCR) in 5- and 10-day-old seedlings and in different *A. thaliana* vegetative and reproductive tissues. *AtZAT4* showed high levels of transcripts in reproductive tissue, specifically in flowers and siliques. The RT-qPCR analysis also revealed moderate expression in stems; however, the transcript levels decreased in young seedlings and roots (Figure 3). These results were in agreement with the expression of *AtZAT4* (AT2G45120) visualized in silico in the eFP-browser (http://bar.utoronto.ca/efp/cgi-bin/efpWeb.cgi, accessed on 21 April 2015), in which *AtZAT4* showed high levels of expression in mature pollen and stamens, and also in seeds.

### 2.2. AtZAT4 Is Essential for the Development of Reproductive Organs in A. thaliana

We used the line CS841944, which has a T-DNA insertion in the *AtZAT4* promoter region (Appendix A), to determine the *AtZAT4* function. The selection of seedlings in growth medium revealed a survival rate of 48.9% instead of the 75% expected from Mendelian inheritance. Surviving seedlings were genotyped to determine the presence of the wild-type and/or the allele containing the T-DNA insertion (Appendix A). The total number of plants analyzed (*n* = 47) showed the presence of both alleles simultaneously, indicating that they corresponded to heterozygotes [*Atzat4* (+/−)] (Appendix A). The absence of homozygous seedlings for the allele containing the insertion suggested that this genotype was not viable. Therefore, these results might be an indicator that insertion in *AtZAT4* confers infertility or that it is lethal at an early embryonic stage.

To analyze the *AtZAT4* relative expression in *Atzat4* (+/−) and WT plants, an RT-qPCR of different vegetative and reproductive organs was performed. The results revealed that *AtZAT4* expression was significantly reduced in flowers and siliques in *Atzat4* (+/−), compared to the expression shown in WT plants (Figure 4). These reductions in the expression of *AtZAT4* observed in the *Atzat4* (+/−) plants suggested a possible role of this gene in the reproductive development of *A. thaliana.*

### 2.3. Atzat4 (+/−) Shows Defects in the Male Germline

To evaluate the function of AtZAT4 in reproductive development, we performed a characterization of the heterozygous mutant *Atzat4* (+/−). We first analyzed the male germline phenotype. The pollen grain viability was determined by using modified Alexander staining [34]. In this method, viable pollen grains show a magenta red coloration of the cytoplasm while aborted pollen shows green cell walls [35,36]. We observed no significant difference in pollen viability between *Atzat4* (+/−) and WT, both showing viability around 100% (*n* = pollen grains from 18 anthers of different flowers of each genotype) (Figure 5A,B). However, pollen tube germination and elongation in vitro revealed significant differences in the mutant *Atzat4* (+/−), compared to WT. In fact, *Atzat4* (+/−) showed reduced germination and smaller tubes in our conditions (Figure 5C–E). *Atzat4* (+/−) presented a 34% (102 ± 3) germination, compared to a 56% (168 ± 4) in WT (Figure 5C). In addition, *Atzat4* (+/−) pollen tubes presented reduced elongation, with 164 ± 6 μm compared to 303 ± 8 μm in WT (Figure 5D,E). These results indicated that AtZAT4 activity was necessary for pollen grain germination and pollen tube elongation.

### 2.4. Atzat4 (+/−) Shows Defects in the Fertility

We further measured *Atzat4* (+/−) fertility in comparison to WT. For this purpose, we decolored siliques and quantified the number of seeds. The results revealed a reduced number of seeds in *Atzat4* (+/−), which was associated with a greater number of unfertilized eggs (Figure 6A,B). Thus, we counted 39 ± 2 seeds per silique on average in WT, while only 27 ± 1 seeds per silique in *Atzat4* (+/−). We also quantified the siliques’ sizes. The siliques of mutants were 12 ± 2 mm in *Atzat4* (+/−), being shorter than WT, which accounted for 14.5 ± 1 mm (Figure 6C). Then, viability and germination capacity assays of seeds were performed. To determine seed viability, the Tetrazolium test was used according to Verma and Majee (2013) [37]. The results showed that the viability of seeds was significantly reduced in the mutant *Atzat4* (+/−) with a total of 60% (60 ± 2) compared to 98% (98 ± 0.04) in WT (Figure 6D,E). These differences seem to be explained by the high number of aborted seeds in the mutant phenotype (Figure 5B). The seed germination was also significantly reduced in *Atzat4* (+/−) with a 49% (49 ± 2) versus 88% (88 ± 4) in WT (Figure 6F). Taken together, these results indicated reduced fertility of *Atzat4* (+/−), suggesting that AtZAT4 was also involved in the development of the seed. Since the expression level of *AtZAT4* in the *Atzat4* (+/−) mutants was insufficient for the correct development of seeds, a high number of ovules were aborted.

## 3. Discussion

Several C_2_H_2_-type ZFPs have been described to participate in anther and pollen development in Arabidopsis and other model plant species [16,20,21,22,23,24]. *AtZAT4* was primarily identified by Mittler et al. (2006) [27] by evaluating the expression of some genes of the *AtZAT* family under different stress conditions, but so far *AtZAT4* from *A. thaliana* has not been characterized in male reproductive and seeds development.

The phylogenetic analysis revealed that AtZAT4 is highly conserved with other C_2_H_2_-ZFPs described in the reproductive development of other plant species (Figure 1 and Appendix A) [29,30]. PEThyZPT3-3 and SlZPT3-1 have three C_2_H_2_-ZF domains, the same as AtZAT4. The function of PEThyZPT3-3 has been associated with reproductive development, where *PEThyZPT3-3* is expressed in stigma, style, ovary, and receptacle of petunia. Interestingly, *PEThyZPT3-3* activity could be related to pollen tube guidance [29]. Moreover, *SlZPT3-1* from *Silene latifolia* is expressed predominantly in male flowers and, to a lesser extent, in female ones [30]. These antecedents allow us to suggest that AtZAT4 could be involved in the reproductive development of Arabidopsis.

AtZAT4 is a C_2_H_2_-type ZFPs with three different types of the conserved plant-specific sequences QALGGH in all ZF domains: the first one is a K2-1 type, the second is a Q2-2 type, and the last one is Q2-3 type (Appendix A). Therefore, it has two invariant Q-type motifs [9]. Additionally, AtZAT4 has an EAR repression motif, which has been associated with repressive activity and is found at the C-terminus of the proteins (Appendix A) [38]. Several studies have shown that C_2_H_2_-ZFPs containing QALGGH and EAR transcriptional repression motifs participate in the development of plant reproductive organs [16,19,22,24]. In addition, according to in silico prediction, AtZAT4 has an NLS (Appendix A). This signal was functional, since the analysis of the subcellular localization showed that AtZAT4 was localized in the nucleus (Figure 2). This suggests that AtZAT4 could act as a transcriptional factor modulating the gene expression related to the reproductive development of *A. thaliana* in a similar way to other previously described C_2_H_2_ ZFPs.

Members of the ZAT protein family have been associated with abiotic stress responses. For instance, AtZAT12 has been involved in response to cold and oxidative stresses [39,40] while AtZAT10 has been associated with plant responses to drought, salinity and cold stresses [27,41]. Furthermore, AtZAT7 participates in tolerance to salt stress [42]. The *AtZAT4* transcriptional profile indicates a higher expression in flowers and siliques (Figure 3), suggesting a role in reproductive organ development. As far as we know, no data is available about the role of the ZAT family in reproductive development. Although our results suggest that AtZAT4 might be involved in male and seed development, its involvement in stress response cannot be ruled out.

An insertional heterozygous T-DNA mutant was used to investigate the function of AtZAT4. Interestingly, in vitro seedling selection revealed that homozygous plants were not viable as all seedlings surviving DL-phosphinothricin herbicide were heterozygous [*Atzat4* (+/−)] (data not shown). This suggested that AtZAT4 was essential for the development of *A. thaliana*. In addition, transcriptional analysis by RT-qPCR revealed the *Atzat4* (+/−) mutant showed a significant reduction in *AtZAT4* expression of flowers and siliques compared to WT (Figure 4). According to eFP-browser 2.0 (http://bar.utoronto.ca/efp/cgi-bin/efpWeb.cgi accessed on 19 November 2020), early stages in embryo and seed development showed high *AtZAT4* expression. This could be related to the lack of viability of embryos homozygous in the insertional mutant, and, therefore, all germinated seedlings showed a WT allele.

The phenotype characterization of the *Atzat4* (+/−) mutant showed similar traits to other male-sterile plants [16,20,23,24]. We observed significative reduced germination and pollen tube elongation in the heterozygous insertional mutant compared with the WT (Figure 5C–E). Both characteristics are essential for the delivery of sperm cells into the ovary, and, thus, for double fertilization [43]. Poor germination and pollen tube elongation may indicate defects during anther and pollen development, as well as during pollen maturation [44]. This suggested that this TF plays an important role during anther and pollen development, and its defect causes male sterility, as other C_2_H_2_-type ZFPs, such as TAZ1 or BcMF20, involved in petunia and *B. campestris* tapetum development, respectively, or in pollen grain wall development as Arabidopsis MAZ1. It has been described that the loss of function of these C_2_H_2_ ZFPs leads to a reduced germination capacity [16,20,23]. On the other hand, Lyu et al. (2020) [24] characterized several C_2_H_2_ ZFPs in *B. rapa* ssp. *chinensis*. One of them is BrZFP38 with two EAR motifs and is functional during male germline development. Interestingly, *BrZFP38* showed high expression in mature pollen grains, but also in the pollen tube during the germination and fertilization processes.

After a double fertilization process, the seed is formed from the ovule, and the fruit is differentiated from the carpel [45,46]. We characterized AtZAT4 in *A. thaliana* fertility. The results evidenced reduced fertility, with a lower number of seeds per silique and smaller silique size, as well as lower seed viability in comparison to WT (Figure 6). Similar to other C_2_H_2_ ZFPs involved in pollen development, these results suggested a role for this TF in *A. thaliana* embryo and seed development. Defects in *DAZ1*/*DAZ2* from *A. thaliana*, which have two EAR motifs, result in abnormal pollen mitosis, and reduced fertility with a lack of embryo and seed development [22]. On the other hand, in petunia, MEZ1 showed incomplete embryo development in addition to defects during the male meiosis [21]. In our research, the TZ test revealed an approximation of the AtZAT4 role in embryo development showing a high number of aborted seeds in the mutant, and, thus, an abnormal embryo development. This could suggest that AtZAT4 is involved in embryo development like other C_2_H_2_-ZFPs. In summary, the high *AtZAT4* expression level in siliques, in addition to the reduced number of seeds per silique, and the lower seed viability and germination rate observed in *Atzat4* (+/−) suggest a putative role during AtZAT4 embryo and seed development. Nevertheless, more studies are needed to confirm this hypothesis.

## 4. Materials and Methods

### 4.1. Plant Material and Growth Conditions

*Arabidopsis thaliana* wild type (WT) plants [ecotype Columbia-0 (Col-0)] and mutants with insertion of T-DNA in the promoter region of *AtZAT4* (CS841944) were obtained from *Arabidopsis Biological Resource Center* (ABRC, https://abrc.osu.edu accessed on 9 July 2015). The plants were grown on a sterile substrate composed of perlite, vermiculite, and peat (1:1:1), and maintained in greenhouse conditions with long daytime photoperiod (16 h light/8 h dark). Plants were fertilized every two weeks with commercial Ultrasol^®^ solution (SQM, Santiago, Chile), and kept until obtaining reproductive structures (flower buds, flowers and siliques) for phenotyping assays, as well as for DNA or RNA extraction. 

### 4.2. Genotyping and Plant Selection

Seeds from *A. thaliana* self-pollinated heterozygous plants *Atzat4* (+/−) (*n* = 96) were germinated on MS (Murashige and Skoog) culture medium with Gamborg vitamins (PhytoTech Labs, Lenexa, KS, USA), supplemented with 100 mg/L of DL-phosphinothricin herbicide (or glufosinate ammonium; PhytoTech Labs, Lenexa, KS, USA), to eliminate WT plants generated during self-pollination and to select those which contained the T-DNA insertion. The surviving plants were analyzed by PCR on gDNA to determine the presence of both the wild-type allele and the T-DNA insertion-containing allele (pCSA110; Appendix A). Seedlings with true leaves (18 day-old) were transplanted into pots containing a mixture of perlite:vermiculite:peat (1:1:1). Arabidopsis *Atzat4* (+/−) plants were selected by application of 120 mg/L of BASTA^®^ 14 SL herbicide (glufosinate ammonium; Bayer, Frankfurt, Germany) and Silwet^®^ L-77 (polyalkyleneoxide modified heptamethyltrisiloxane; Momentive Inc., Waterford, NY, USA). Three applications were made every two days by spraying, along with an application of herbicide to WT plants as a control. The presence of T-DNA in mutant plants was analyzed by PCR on gDNA using specific primers for the insertion (Appendix A). 

### 4.3. RNA Isolation and cDNA Synthesis

For RNA extraction, the samples were immediately frozen in liquid nitrogen and stored at −80 °C. Total RNA was extracted from different tissues of *A. thaliana* and also from 5- and 10-day-old seedlings, using the SV Total RNA Isolation System kit (Promega, Madison, WI, USA). From this, RNA integrity was visualized by 2% (*w*/*v*) agarose gel electrophoresis, and RNA concentration and purity (OD_260_/OD_280_ ratio > 1.95) were determined with an Infinite^®^ 200 PRO NanoQuant (Tecan, Männedorf, Switzerland). All RNA samples were treated with Turbo™ DNase (Invitrogen, Waltham, MA, USA) to remove contaminant DNA traces. To prepare first-strand cDNA, 2 μg of total RNA were reverse transcribed in a 20 μL reaction using the oligo d(T) and AffinityScript qPCR cDNA Synthesis Kit (Agilent Technologies, Inc., Santa Clara, CA, USA), following manufacturer’s instructions. 

### 4.4. Subcellular Localization of the AtZAT4-GFP Fusion Protein 

First, in silico prediction of a Nuclear Localization Signal (NLS) was performed by NLStradamus online software (http://www.moseslab.csb.utoronto.ca/NLStradamus/ accessed on 21 April 2015) [47]. Subsequently, the cDNA of *AtZAT4* was cloned in pGEM^®^-T Vector Easy (Promega, Madison, WI, USA) without stop-codon and with a cleavage site for *Nco*I at its 3′ end. The resulting plasmid was treated with *Nco*I enzyme and the released fragment was ligated into the pGFPau vector, containing GFP, under the control of the Cauliflower Mosaic Virus (CaMV) 35S promoter. The constructs were transferred into onion epidermal cells by particle bombardment using a Bio-Rad Biolistic PDS 1000/He system. The samples were incubated for 24 h at 22 °C in darkness. The protein localization was determined by a Leica SP2 confocal microscope (Leica, Wetzlar, Germany). 

### 4.5. Analysis of Gene Expression

The relative *AtZAT4* expression was analyzed by reverse transcription-quantitative real-time PCR (RT-qPCR) in different tissues of *A. thaliana* and in 5- and 10-day-old seedlings of WT and *Atzat4* (+/−) using a Stratagene Mx3000P QPCR System (Agilent Technologies, Inc., Santa Clara, CA, USA). The 2X Maxima^®^ SYBR Green/ROX qPCR Master Mix (Thermo Scientific, Waltham, MA, USA) was used in all reactions, according to the protocol described by the manufacturer. For each of the three biological replicates used, RT-qPCR was carried out in triplicate, using 10 μL of Master Mix, 0.5 μL of 250 nM primers, 2 μL of cDNA (25 ng/ μL), and nuclease-free water to a final volume of 20 μL. Amplification was followed by a melting curve analysis with continuous fluorescence acquisition during the 55–95 °C melt. The gene expression was normalized against *AtF-box* (AT5G15710) [32], and the fold change of gene transcript levels was calculated using the 2^−ΔΔCt^ method [33]. 

### 4.6. Construction and Analysis of Phylogenetic Tree 

C_2_H_2_ ZFP sequences were obtained from the NCBI database (https://www.ncbi.nlm.nih.gov/ accessed on 6 April 2015). Multiple protein sequence alignments were done by using the MEGA X software (version 11.0) [48] with ClustalW, and default parameters [49]. For the alignment edition, we used BioEdit (version 7.2) [50]. The identification of the C_2_H_2_ Zinc finger domains was performed in InterProScan (https://www.ebi.ac.uk/interpro/search/sequence/ accessed on 21 April 2015) [51]. The nuclear localization signal was determined in NLStradamus (http://www.moseslab.csb.utoronto.ca/NLStradamus/ accessed on 21 April 2015) [47], and the EAR motif was determined literature-based [52]. On the other hand, phylogenetic analysis was performed with MEGA X software (version 11.0 Philadelphia, USA) [48] by the Neighbor-Joining method [53], and the bootstrap resampling method using 1000 replicates [54]. The GenBank accession numbers of plant ZFPs used for phylogenetic tree are the following: *Arabidopsis thaliana* AtZAT4 (AT2G45120), AtMAZ1 (AT5G15480), AtDAZ1 (AT2G17180), AtDAZ2 (AT4G35280); *Petunia hybrida* PEThyZPT3-3 (BAA96071), PEThyZPT4-2 (BAA19926), PEThyZPT4-3 (BAA20137), PEThyTAZ1 (BAA19113), PEThyZPT4-1 (BAA19114), PEThyMEZ1 (BAA19110); *Silene latifolia* SlZPT3-1 (AAY40249); *Brassica campestris* ssp. *chinensis* (ADK92391). The sequence of BrZFP38 from *Brassica rapa* ssp. *chinensis* (Bra011631) was obtained from BRAD (http://brassicadb.cn/ accessed on 5 January 2021). 

### 4.7. Pollen Viability

For the pollen grain viability analysis, modified Alexander staining was performed [34]. Arabidopsis WT and mutant *Atzat4* (+/−) flowers in stage 12 of flower development [55,56] from three different plants were dissected under the Olympus SZ30 magnifier (Tokyo, Japan) to obtain complete stamens. They were fixed and discolored in Carnoy solution, with absolute ethanol, chloroform, and acetic acid (6:3:1) for at least 2 h. The fixed stamens were placed on a slide with 2-4 drops of modified Alexander solution, following the described protocol [34]. Subsequently, the staining was visualized under the optical microscope (Wild Leitz GMBH, Germany). Later, the number of viable pollen grains from the anthers of three biological replicas was quantified, thus obtaining the rate of viability for each genotype.

### 4.8. In Vitro Pollen Tube Germination and Elongation 

For the pollen tube germination and elongation analysis, four dehiscent anthers (anthesis) of Arabidopsis from WT and *Atzat4* (+/−) from three different plants were used. The pollen grains of each flower were deposited on slides with a solid germination medium with 200 µL of 500 mM KCl, 200 µL of 500 mM CaCl_2_, 200 µL of 100 mM MgSO_4_, 200 µL of 1% (*w*/*v*) H_3_BO_3_, 4 g sucrose and 0.3 g agarose [1.5% (*w*/*v*)] for 20 mL of the final solution, pH 7.5. The samples were covered with a dialysis membrane and were kept in a humid chamber for 6 h at 28 °C. The pollen germination and elongation were visualized under a 10X lens in the Zeiss LSM700 microscope in brightfield and the image analysis was performed using ZEN-Zeiss software (Oberkochen, Germany). Finally, the germinated pollen grains quantification and the length of the pollen tubes measurement were performed in ImageJ software (National Institute of Health, Bethesda, Maryland, USA). Using these results, the pollen grain germination percentage and pollen tube size average were calculated.

### 4.9. Seeds per Siliques and Silique Size Quantification 

Discoloration of mature, non-dehiscent silique was performed for WT and *Atzat4* (+/−); The seeds quantification was performed under the Olympus SZ30 magnifying glass (Tokyo, Japan). Seeds of 10 siliques from three different plants were quantified. Furthermore, the sizes of 12 mature non-dehiscent siliques from three different plants were quantified for the two genotypes. 

### 4.10. Seed Viability and Germination

Seed viability analysis was performed using the tetrazolium test (2,3,5-triphenyl tetrazolium chloride; TZ) for WT and *Atzat4* (+/−) plants. TZ precipitates to 2,3,5-triphenyl formazan red by the activity of dehydrogenases present in living cells [37,57,58]. One hundred seeds were taken in three technical replicates, with the protocol described by Verma & Majee (2013) [37]. Finally, the quantification of viable, non-viable and defective seeds was carried out under the Olympus SZ30 magnifying glass (Tokyo, Japan). At the same time, a germination analysis of Arabidopsis seeds for two genotypes was performed in an MS culture medium with Gamborg vitamins (PhytoTech Labs, Lenexa, KS, USA) and 3% (*w*/*v*) sucrose. We used ~50 seeds in triplicate, which were disinfected. Subsequently, the seeds were placed in a growth chamber, and the germinated seeds were quantified on the third day of growth.

### 4.11. Statistical Analysis 

Statistical analyses were performed by using the software R and Rcmdr package (http://knuth.uca.es/R/doku.php?id=instalacion_de_r_y_rcmdr:r-uca accessed on 10 January 2018) (John Fox, Hamilton, ON, Canada). The analyses included T-Student for phenotypic characterization and one-way ANOVA for relative expression data using Tukey test at *p* ≤ 0.05 to determine significant differences between means. The results were expressed as means ± standard error (SE).

## 5. Conclusions

In conclusion, we have characterized AtZAT4, for the first time, at the functional level in the *A. thaliana* reproductive development. The *Atzat4* (+/−) insertional mutant revealed a reduced expression in flowers and siliques, and the phenotypic characterization suggested a reduced activity of this transcription factor, which resulted in defects in male germline and fertility of *A. thaliana*. *Atzat4* (+/−) presented reduced germination and pollen tube elongation, as well as a lower number of seeds per silique and a reduced fruit size. In addition, *Atzat4* (+/−) revealed lower viability and seed germination. Based on these antecedents, we suggest that AtZAT4 regulates the expression of genes involved in pollen development, and is likely to also regulate genes implicated in embryo and seed development.

## Figures and Tables

**Figure 1 plants-11-01974-f001:**
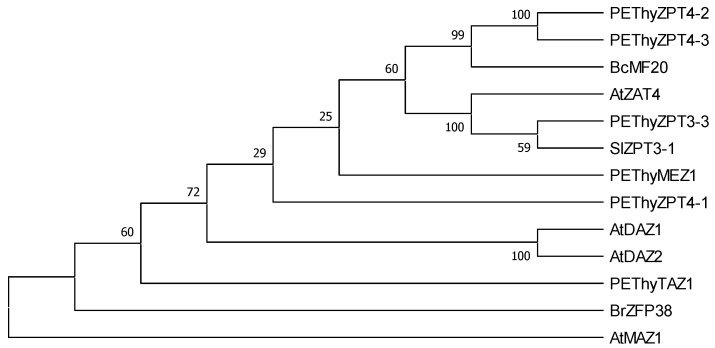
Molecular phylogenetic tree of AtZAT4 with other C_2_H_2_-Zinc Finger Proteins (C_2_H_2_-ZFPs). The tree was obtained from the multiple alignments of the deduced AtZAT4 and other C_2_H_2_-ZFPs following the Neighbor-Joining method in the MEGA-X package with 1000 replicates for bootstrap values. For details, see Section 4.

**Figure 2 plants-11-01974-f002:**
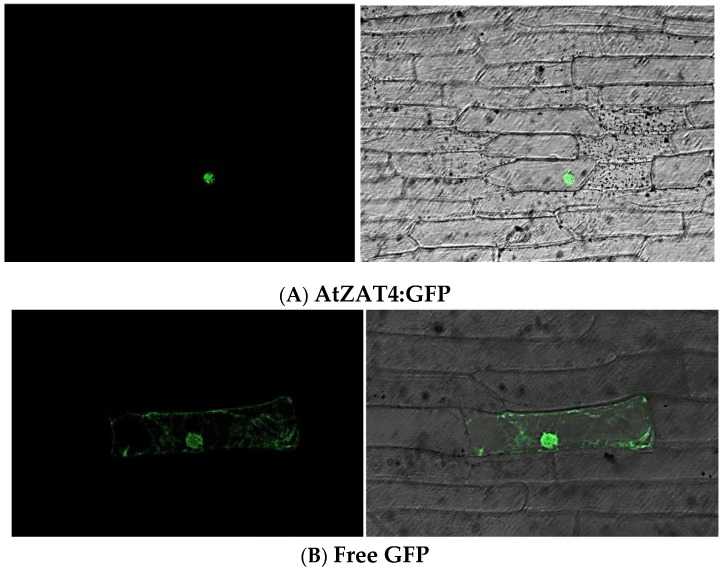
Subcellular localization of the AtZAT4 protein. (**A**) AtZAT4:GFP or (**B**) constructs carrying GFP were bombarded into onion epidermal cells. GFP and AtZAT4:GFP fusion proteins were under the control of the CaMV 35S promoter. After the bombardment, the samples were incubated for 24 h at 22 °C in the dark (left column) and then visualized by a confocal microscope (right column). Each experiment was done in triplicate resulting in the same fluorescence pattern. Bar = 100 µm. For details, see Section 4.

**Figure 3 plants-11-01974-f003:**
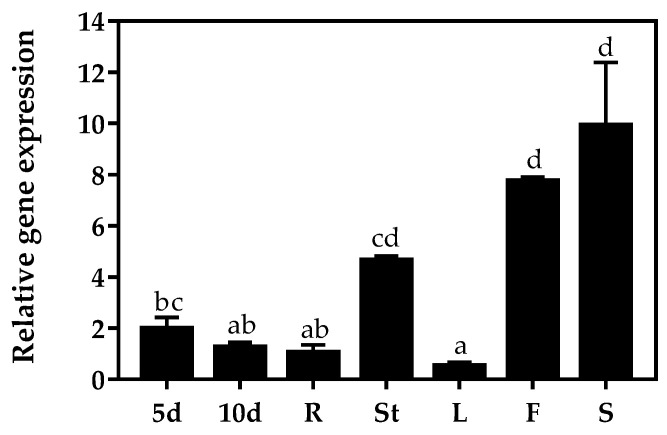
Gene expression profile of *AtZAT4* in vegetative and reproductive tissues of *Arabidopsis thaliana* and in 5- and 10-day-old seedlings. **5d** seedlings of 5-day-old, **10d** seedlings of 10-day-old, **R** root, **St** stem, **L** leaf, **F** flower, **S** silique. Values represent mean ± SE (*n* = 3). The constitutive expression of the gene *AtF-box* [32] and the 2^−ΔΔCt^ method described by Livak and Schmittgen (2001) [33] were used for normalization. The transcript levels obtained for R were taken to assign the value one. Data correspond to the mean ± SE. Significant differences (*p* ≤ 0.05) are shown with different letters. For details, see Section 4.

**Figure 4 plants-11-01974-f004:**
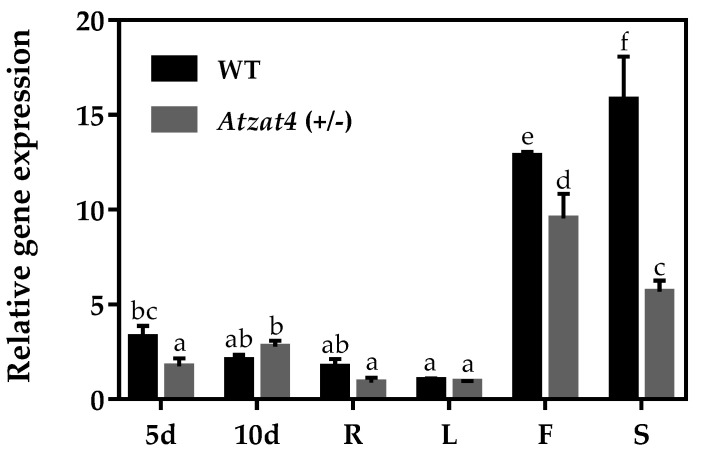
Gene expression profile of *Atzat4* (+/−) in vegetative and reproductive tissues of *Arabidopsis thaliana* and in 5- and 10-day-old seedlings. **5d** Five-day-old seedlings, **10d** Ten-day-old seedlings, **R** root, **L** leaf, **F** flower, **S** silique. Values represent mean ± SE (*n* = 3). The constitutive expression of the gene *AtF-box* [32] and the 2^−ΔΔCt^ method described by Livak and Schmittgen (2001) [33] were used for normalization. The transcript levels obtained for R were taken to assign the value one. Data correspond to the mean ± SE. Significant differences (*p* ≤ 0.05) are shown with different letters. For details, see Section 4.

**Figure 5 plants-11-01974-f005:**
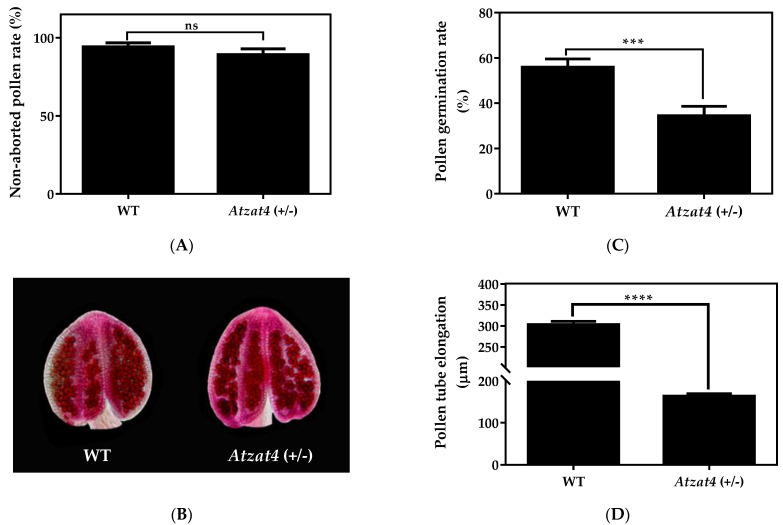
Male germline in *Atzat4* (+/−) compared to WT in *Arabidopsis thaliana*. (**A**,**B**) Modified Alexander stain [34]. (**A**) Non-aborted pollen grain quantification. (**B**)**.** Alexander red staining of pollen grains by anther, non-aborted pollen grains are stained red. (**C**–**E**) In vitro pollen grain germination and pollen tube elongation*. (***C**) Quantification of germinated pollen grains (*n =* 300 pollen grains analyzed). (**D**) Pollen tube elongation quantification (*n* = 300 pollen grains germinated). (**E**) In vitro pollen grain germination and pollen tube elongation under the light microscope. In (**A**,**C**,**D**) data correspond to the mean ± SE; ******** *p* < 0.0001, *******
*p* < 0.001, **ns:** not significant. In (**B**,**E**), bars = 100 µm. For details, see Section 4.

**Figure 6 plants-11-01974-f006:**
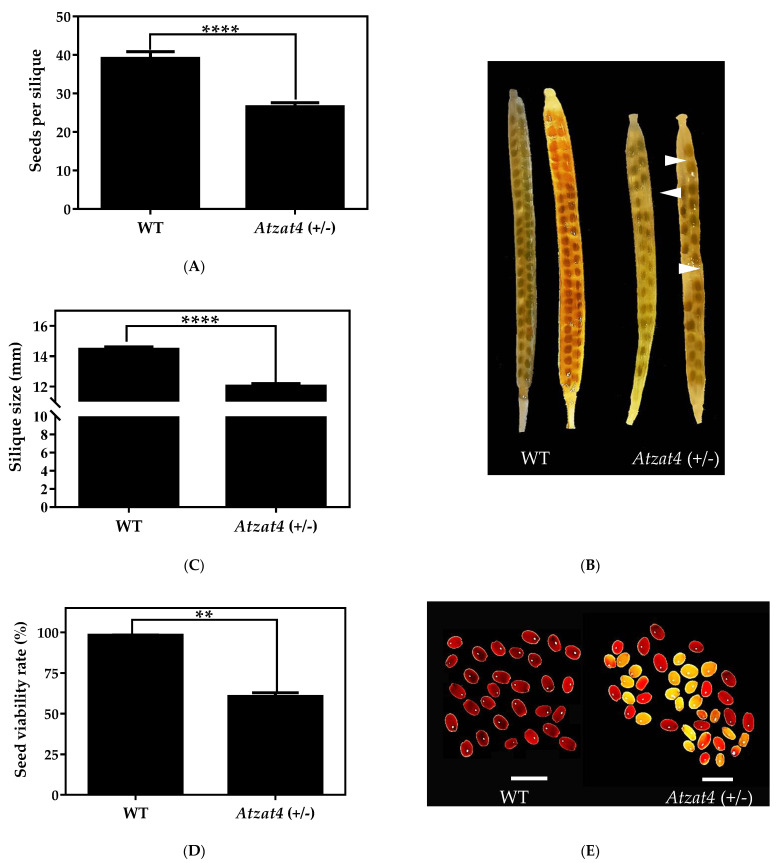
Fertility in *Atzat4* (+/−) compared to WT in *Arabidopsis thaliana*. (**A**) Seeds per silique quantification (*n* = 10 siliques in triplicate). (**B**) Discolored siliques with developed seeds and unfertilized ovules (white arrowheads). (**C**) Silique size quantification (*n* = 10 siliques in triplicate). (**D**) Viable seeds quantification using Tetrazolium test (*n* = 100 seeds in triplicate). (**E**) Tetrazolium test, red staining shows viable seeds, unstained unviable seeds and pink stain indicates dead tissue. (**F**) Quantification of germinated seeds (*n* = 100 seeds in triplicate). In (**A**,**C**,**D**,**F**) data correspond to the mean ± SE; ******** *p* < 0.0001, ****** *p* < 0.01, ***** *p* < 0.05. In (**B**,**E**), bars = 1 mm. For details, see Section 4.

## Data Availability

Not applicable.

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
