# Peer review of "AtZAT4, a C2H2-Type Zinc Finger Transcription Factor from Arabidopsis thaliana, Is Involved in Pollen and Seed Development"

_plants, 2022, doi:10.3390/plants11151974_

Round 1

Reviewer 1 Report

Please see my comments in the attached file.

Author Response

Dear Reviewer;

1.- All spelling or formatting errors noted by the reviewer have been corrected in the text of the new manuscript version.

2.- Why this ratio1:1:1?  line 405

Response:

The substrates must provide the plants with aeration, support and hydration. The 1:1:1 ratio of perlite/vermiculite/peat provides those suitable conditions for Arabidopsis thaliana’s growth allowing to establish reproducible conditions as has been widely reported by several authors (e.g. Rock & Zeevart, 1991, PNAS, https://doi.org/10.1073/pnas.88.17.7496)

3.- Why these conditions? Line 406

Response:

These long-day photoperiod conditions in Arabidopsis thaliana induce flowering. Since this research analyzes the reproductive process, both mutant and WT (Col-0) plants were subjected to these growth conditions (these photoperiod conditions have been widely reported for flowering induction in Arabidopsis).

4.- Why after 18 days? Line 421

Response:

After 18 days, the plants showed true leaves and the PCR selection assay using gDNA was performed.

The text was changed as follows: “Seedlings with true leaves (18 day-old) were transplanted….”

5.- How you carried out the spraying? Line 425

Response:

Spraying was carried out with a sprayer containing 120 mg/L BASTA on potted seedlings.

In addition, the manuscript with the answers is attached.

Reviewer 2 Report

In the manuscript, the authors provided a series of data to discover the function of a transcription factor, AtZAT4, in pollen development and fertilization. The story is novel and interesting. The figures are clear. The conclusion is logic and reliable. Nevertheless, the results look a little preliminary to build a good story. To my opinion, the authors should provide more data. Here I have two recommendations:

(1) Backcrossing. Pollinate atzat4 pollen onto the stigma of wildtype, pollinate wildtype pollen onto stigma of atzat4, then investigate in vivo pollen germination & tube growth (by using Aniline blue staining) and the development of siliques and seeds. It will provide evidence to verify AtZAT4 function in pollen germination, tube growth or in pistil development & pollen acceptance, or both of them.

(2) Screening genes whose expression may be regulated by AtZAT4. To verify the function of a transcription factor, such an investigation is necessary and helpful. I recommend the authors to do chromatin immunoprecipitation and sequencing the enriched genes. 

Author Response

Dear Reviewer;

Suggestion 1

Backcrossing. Pollinate atzat4 pollen onto the stigma of wildtype, pollinate wildtype pollen onto stigma of atzat4, then investigate in vivo pollen germination & tube growth (by using Aniline blue staining) and the development of siliques and seeds. It will provide evidence to verify AtZAT4 function in pollen germination, tube growth or in pistil development & pollen acceptance, or both of them.

Response:

We appreciate the reviewer's suggestion. However, the Atzat4 (+/-) phenotype generates both pollen with and without the ability to germinate, so backcrossing with Atzat4 (+/-) pollen onto stigma of wildtype to investigate germination and tube growth in vivo would be very difficult to realize. In this sense, we already showed by in vitro assays that the gemination rate and pollen tube elongation of Atzat4 (+/-) is reduced in comparison with WT (Figure 5). The second backcrossing of wildtype pollen onto stigma of Atzat4 (+/-) is feasible to study the level of pollen acceptance of the pistil. However, we think that it could be carried out in future research and would not directly contribute to the conclusions of the present study.

Suggestion 2

  • To verify the function of a transcription factor, such an investigation is necessary and helpful. I recommend the authors to do chromatin immunoprecipitation and sequencing the enriched genes. 

Response:

Thank for this comment. We think that this suggestion is precisely the work that, based on the results shown in the current article, should be carried out to continue with the investigation to be presented in a future publication.

Reviewer 3 Report

The manuscript of Puentes-Romero et al. “AtZAT4, a C2H2-type zinc finger transcription factor from Arabidopsis thaliana, is involved in pollen and seed development” deals with the investigation of the role of AtZAT4 gene protein in fertilization and seed viability in Arabidopsis thaliana. Based on the obtained results, the authors conclude that transcription factor AtZAT4 plays a role in pollen and seed development in Arabidopsis.

The manuscript contains new and valuable information and it is very well and clearly written. The data contribute to revealing the mechanism of regulation of fertility in Arabidopsis thaliana, and thus deserve publication. I have just some minor comments and suggestions to the manuscript.

Methods

Line 409-410 – last sentence in paragraph 4.1 should be moved as the first sentence to the paragraph 4.3.

References

In references no. 25, 48, 49 and 54 the names of the journal are missing - J Exp Bot, Mol Biol Evol, Nucleic Acids Res and Evolution, respectively.

Author Response

Dear Reviewer;

Methods

Line 409-410 – last sentence in paragraph 4.1 should be moved as the first sentence to the paragraph 4.3.

Response;

The change of location of the paragraph has been made in the new versio of  the manuscript 

References

In references no. 25, 48, 49 and 54 the names of the journal are missing - J Exp Bot, Mol Biol Evol, Nucleic Acids Res and Evolution, respectively.

Response;

All references errors has been corrected in the new version of article.

Round 2

Reviewer 2 Report

The revision and response are acceptable.